# High Expression of MRPL23 Is Associated with Poor Survival in Clear-Cell Renal Cell Carcinoma

**DOI:** 10.3390/cancers16233909

**Published:** 2024-11-21

**Authors:** Edyta Podemska, Jędrzej Borowczak, Damian Łukasik, Dariusz Grzanka, Justyna Durślewicz

**Affiliations:** 1Department of Clinical Pathomorphology, Faculty of Medicine, Collegium Medicum in Bydgoszcz, Nicolaus Copernicus University in Torun, 85-094 Bydgoszcz, Poland; 306228@stud.umk.pl (E.P.); damian.lukasik@cm.umk.pl (D.Ł.); d_grzanka@cm.umk.pl (D.G.); 2Department of Oncology and Brachytherapy, Faculty of Medicine, Collegium Medicum in Bydgoszcz, Nicolaus Copernicus University in Torun, 85-796 Bydgoszcz, Poland; jedrzej.borowczak@cm.umk.pl; 3Clinical Department of Oncology, Franciszek Łukaszczyk Oncology Centre, 85-796 Bydgoszcz, Poland; 4Department of Tumor Pathology, Franciszek Łukaszczyk Oncology Center, 85-796 Bydgoszcz, Poland

**Keywords:** MRPL23, clear-cell renal cell carcinoma, immunohistochemistry, patient survival, prognostic biomarker

## Abstract

This study examines the role of MRPL23, a cellular protein, in clear-cell renal cell carcinoma (ccRCC) by comparing its levels in cancerous tissues to those in nearby healthy tissues. Our findings show that MRPL23 protein expression is reduced in tumor tissues, while its mRNA levels are increased, linking it to specific cancer characteristics and progression. Patients with higher MRPL23 protein and mRNA levels had poorer survival outcomes, suggesting that MRPL23 may be a valuable marker for predicting prognosis in ccRCC. These results highlight the potential of MRPL23 to inform treatment strategies and support further research into its role in ccRCC development.

## 1. Introduction

Renal cell carcinoma (RCC) is a malignancy of the urinary system, known for its increasing incidence and high mortality rate. Clear-cell renal cell carcinoma (ccRCC), the most prevalent subtype, represents over 80% of RCC cases [1,2]. One major challenge is the absence of specific clinical symptoms in the early stages, which, along with the lack of reliable diagnostic markers, results in about 30% of ccRCC patients already having metastatic disease at the time of diagnosis [3]. The prognosis for metastatic ccRCC is poor, with a median survival of around 13 months and only about 10% of patients surviving beyond five years [4]. For this reason, discovering novel diagnostic markers, prognostic indicators, and therapeutic targets is essential to improving patient outcomes in ccRCC.

Long non-coding RNAs (lncRNAs) constitute a significant portion of the human genome and play a crucial role in gene regulation, even though they do not encode proteins themselves. These lncRNAs, defined by a length of at least 200 nucleotides, are increasingly recognized for their importance in cancer biology. They influence cancer development and metastasis across various tumor types, including lung cancer, osteosarcoma, and adenoid cystic carcinoma, by modulating microRNA (miRNA) interactions and disrupting the balance of the lncRNA/miRNA/mRNA axis [5,6]. One such lncRNA, MRPL23-AS1, has attracted attention due to its role in cancer. Predominantly localized in the nucleus, MRPL23-AS1 enhances the binding of EZH2 and H3K27me3 to the E-cadherin promoter region, leading to its silencing. This silencing is associated with increased vascular permeability, enhanced metastasis, and accelerated tumor growth [6]. In contrast, MRPL23 is a protein-coding gene that belongs to the group of mitochondrial ribosomal proteins (MRP). MRPs are a component of the mitochondrial ribosome, classified into two subcategories: MRPL (the components of the large ribosomal subunit) and MRPS (the components of the small ribosomal subunits) [7]. These proteins are involved in energy production and cellular metabolism, making it an essential element of cellular function. Out of 80 known MRP genes, several have been studied in various malignancies [8,9,10,11,12,13,14,15]. In the context of cancer, MRPL23 may influence cellular pathways related to stress response and the regulation of cell growth [16]. Although MRPL23 has not been studied as extensively as MRPL23-AS1 in the context of gene regulation, its expression and potential functions in various cancers, including ccRCC, warrant further investigation.

Given the poor prognosis for metastatic ccRCC and the urgent need for new diagnostic and prognostic markers, this study aims to investigate the potential role of MRPL23 in ccRCC. The hypothesis is that MRPL23 expression may have prognostic value in ccRCC. To test this hypothesis, we will analyze MRPL23 expression levels in ccRCC using both proprietary and public datasets, focusing on their impact on clinicopathological features and overall survival (OS) in ccRCC patients [17].

## 2. Materials and Methods

### 2.1. Patients and Tissue Specimens

In the initial screening phase, a total of 132 patients who underwent surgery at the Department of Urology and Andrology, Antoni Jurasz University Hospital No. 1, Bydgoszcz, Poland, were considered for inclusion in the study. To simplify and optimize the study, only cases of clear-cell renal cell carcinoma (ccRCC) were included in the analysis, while other histological types were excluded from the study. Histopathological assessment of each tumor sample was conducted independently by two pathologists to select a representative study cohort at the Department of Clinical Pathomorphology, Collegium Medicum in Bydgoszcz, Nicolaus Copernicus University in Toruń. Ultimately, 99 cases of ccRCC were included in the study. The study group consisted of 68 males and 31 females, with a median age of 64 years (range: 42–83). Histological analysis identified 25 cases of well-differentiated ccRCC, 64 cases of moderately differentiated ccRCC, and 10 cases of poorly differentiated ccRCC. Post-surgery survival data were obtained for all patients, with a median follow-up period of 47 months [18,19].

The study protocol received approval from the Ethics Committee of Nicolaus Copernicus University in Toruń, Ludwik Rydygier Collegium Medicum in Bydgoszcz (approval number KB 253/2018). All methodologies adhered to the principles of good laboratory practice.

### 2.2. Tissue Microarrays (TMA) and Immunohistochemical Staining

Immunohistochemical staining (IHC) was performed on tissue macroarrays containing representative tumor areas, where five different tissue fragments from donor FFPE (formalin-fixed, paraffin-embedded) blocks were placed in a single recipient block. Sections 4 μm thick were cut from each tissue macroarray block and placed on high-adhesion glass slides (SuperFrost Plus; Menzel-Glaser, Braunschweig, Germany). The immunohistochemistry staining was conducted using the BenchMark^®^ Ultra automated staining device (Roche Diagnostics/Ventana Medical Systems, Tucson, AZ, USA) with the Ventana UltraView DAB Detection Kit (Ventana Medical Systems) according to previously described procedures. MRPL23 was detected using a rabbit polyclonal anti-MRPL23 antibody (cat. no.: HPA050406, Sigma-Aldrich, St. Louis, MO, USA) at a 1:100 dilution.

### 2.3. Evaluation of Immunohistochemistry Staining

Protein expression assessment via IHC staining was carried out on tumor tissues and adjacent control tissues. Slide images were captured using a Roche Ventana DP 200 scanner and evaluated by both an image scientist and a pathologist. Immunoreactivity analysis utilized a customized Index Remmele–Stegner scale (IRS), encompassing a range from 0 to 12. This scale integrates the percentage of positively stained cells (0–4) with staining intensity (0–3). The results of IHC analysis for MRPL23 were categorized as follows: 0 (no positive cells), 1 (less than 10% of cells), 2 (10–50% of stained cells), 3 (51–80% of stained cells), and 4 (more than 80% of stained cells). Intensity scoring ranged from 0 (negative) to 3 (strong staining). The findings of the evaluated proteins were stratified into groups indicating low and high expression levels using the Evaluate Cutpoints program based on the optimal cut-off value [20]. To define the levels of MRPL23 expression, a median-based threshold was established: values below 7 signify low expression, while values of 7 or higher indicate high expression.

### 2.4. In Silico Analysis of TCGA Data

Survival and gene expression data for a cohort of 475 patients with ccRCC from The Cancer Genome Atlas (TCGA) were obtained from the UCSC Xena Browser (http://xena.ucsc.edu/; accessed on 6 July 2024) [21]. Gene expression data from the Genotype-Tissue Expression (GTEx) project were also included for comparison. The differential expression analysis was performed using DESeq2, and the data were expressed as RSEM expected_count (DESeq2 standardized). The data were divided into low and high expression groups according to the cut-off points provided by the Evaluate Cutpoints software. The cut-off values for MRPL23 expression were as follows: <10.84 for low expression and ≥10.84 for high expression.

### 2.5. Statistical Analysis

Statistical analyses were performed using GraphPad Prism software version 10.1 (GraphPad Software, San Diego, CA, USA) and SPSS version 29.0 (IBM Corporation, Armonk, NY, USA). The normality of the data distribution was assessed using the Shapiro–Wilk test. Continuous variables were compared using the Mann–Whitney test, while the significance of clinical factors was assessed using the chi-square test or Fisher’s exact test. Survival analysis was conducted using the Kaplan–Meier method, and differences between groups were evaluated using the log-rank test. Both univariate and multivariate Cox proportional hazards regression analyses were performed, and hazard ratios (HRs) along with 95% confidence intervals (95% CI) were also estimated. Statistical significance was set at *p* < 0.05.

## 3. Results

### 3.1. Assessment of MRPL23 Protein Expression in ccRCC and Adjacent Non-Tumorous Tissues Using Immunohistochemistry

The expression of MRPL23 protein in our cohort was assessed using IHC in 99 ccRCC and 30 non-tumorous adjacent tissues. MRPL23 IHC staining was detected in the cytoplasmic compartments of ccRCC cells. According to the established cut-off point, a high cytoplasmic immunoreactivity of MRPL23 was observed in 50 (50.51%) ccRCC cases, while the remaining 49 (49.49%) exhibited low expression. Representative images illustrating MRPL23 expression are shown in Figure 1A–C. As demonstrated by the conducted analyses, shown in Figure 2, MRPL23 expression was reduced in cancerous epithelial cells of ccRCC tissues compared to renal tubular epithelial cells in normal tissues (*p* ≤ 0.05; Figure 2A). No relationships were observed between clinicopathologic characteristics and MRPL23 protein expression (Table 1).

### 3.2. Assessment of MRPL23 mRNA Expression in ccRCC and Adjacent Non-Tumorous Tissues Using TCGA Data

Based on the cut-off values determined for MRPL23 mRNA, high expression was observed in 240 (50.53%) ccRCC cases, while low expression was found in 235 (49.47%) cases. In the analysis of the TCGA cohort, MRPL23 mRNA levels were significantly elevated in tumor tissues compared to normal non-tumorous tissues (*p* ≤ 0.0001; Figure 2B). The analysis of the correlation between MRPL23 expression and clinicopathological characteristics revealed significant differences between MRPL23 expression status and gender, tumor grade, pT status, and disease stage (Table 2). MRPL23 expression is statistically significantly different based on gender (*p* = 0.0203). MRPL23 expression increases with the increase in tumor grade (*p* = 0.0079). MRPL23 expression is higher in more advanced pT stages (*p* < 0.0001). MRPL23 expression increases with the progression of the disease stage (*p* < 0.0001).

### 3.3. Survival Outcomes Based on Protein Expression Levels of MRPL23 in ccRCC Patients

Kaplan–Meier survival curves demonstrated that OS was significantly worse in ccRCC patients with high MRPL23 protein expression compared to those with low expression (median OS of 28 and 61 months, respectively; *p* = 0.005, Figure 2C). A univariate Cox analysis showed that high MRPL23 protein expression was significantly associated with a worse survival prognosis (HR 1.81, 95% CI 1.18–2.77, *p* = 0.01; Table 3), and it remained an independent prognostic factor for deteriorations in OS in a multivariate Cox analysis after adjustment for gender, age, grade, and cN (HR 1.66, 95% CI 1.07–2.56, *p* = 0.02; Table 3).

### 3.4. Survival Outcomes Based on mRNA Expression Levels of MRPL23 in ccRCC Patients

Kaplan–Meier survival curves revealed that OS was significantly shorter in ccRCC patients with high MRPL23 protein expression compared to those with low expression (median OS of 75 months versus not available; *p* = 0.006, Figure 2D). A univariate Cox analysis showed that MRPL23 mRNA expression was significantly associated with worse overall survival (HR = 1.56, 95% CI 1.13–2.15; *p* = 0.01; Table 4). However, in a multivariate Cox analysis, MRPL23 mRNA expression was found not to be an independent adverse prognostic factor for OS after adjusting for other variables (HR = 1.18, 95% CI 0.85–1.64; *p* = 0.32; Table 4).

## 4. Discussion

The contemporary advancements in biological sciences and technological progress present immense opportunities, offering hope for understanding the molecular mechanisms of disease development and for the creation of modern treatment methods. Although the literature is rich with descriptions of recent trends, studies focusing on protein-coding genes, such as MRPL23, hold significant importance in cancer research and clinical practice. MRPL23, as a key component of the mitochondrial ribosome, plays a crucial role in mitochondrial protein synthesis and is involved in energy production and cellular metabolism. Alterations in the function or expression of mitochondrial proteins can disrupt cellular homeostasis, potentially promoting tumorigenesis [16].

In our institutional cohort, we demonstrated that MRPL23 expression was reduced in ccRCC compared to normal tissues. In contrast, the analysis of the TCGA cohort showed that MRPL23 mRNA levels were significantly elevated in tumor tissues compared to non-tumorous tissues. These differences may result from various levels of gene expression regulation, including both post-transcriptional and translational regulation, which can affect mRNA stability as well as protein production and degradation. Differences may also stem from the variability of the tumor microenvironment, where different cell subpopulations can exhibit distinct expression profiles at the mRNA and protein levels. No associations were observed between MRPL23 protein expression and clinicopathological characteristics in our institutional cohort. However, the analysis of TCGA data revealed significant correlations between MRPL23 mRNA expression and clinicopathological features. These findings suggest that elevated MRPL23 expression may contribute to tumor aggressiveness and progression. The positive correlation with higher tumor grades and advanced pT stages further supports the potential role of MRPL23 in promoting tumor growth and progression. Kaplan–Meier survival curves demonstrated that OS was significantly worse in ccRCC patients with high MRPL23 protein and mRNA expression. Furthermore, MRPL23 protein expression remained an independent prognostic factor for poorer OS in a multivariate Cox analysis. This may indicate the significant role that MRPL23 plays in tumor biology, particularly in the context of its impact on cellular metabolism and signaling pathways related to cell proliferation and oxidative stress response.

To date, the role of MPRL23 in ccRCC remains unexplored. The MPRL23 gene is located on human chromosome 11p15.5 and is bilaterally expressed despite its location in a region of imprinted genes. MRPL23 may be present in both the nucleus and cytoplasm [16]. Early studies indicate that MRPL23 is a structural component of a large ribosomal subunit involved in mitochondrial translation [16]. Thus, alterations in MRPL23 expression levels may be associated with carcinogenesis and enhanced tumor progression. Our previous study determined that high MPRL23 expression was associated with a higher risk of unfavorable outcomes in ccRCC patients [19]. Furthermore, the MRPL23 activity was intertwined with the signaling network, including genes that encode adenine phosphoribosyltransferase (APRT), which catalyzes the formation of AMP; mitochondrial protein MPRS15, which builds the 28S subunit protein; and 40S ribosomal protein S15 (PRS15), whose mutations repress RNA translation in cancers [21,22]. Although there is a limited number of studies on MRPL23 and its role in cancer, significantly more attention has been given to MRPL23-AS1, which has been associated with cancer processes and gene regulation [3,4]. It is also worth emphasizing that the presence of both MRPL23 and MRPL23-AS1 on the chromosome 11p15.5 suggests a potential interdependence of their functions in regulating cancer processes. While MRPL23 plays a crucial role as a mitochondrial protein involved in protein synthesis and cellular metabolism, MRPL23-AS1 acts as an lncRNA. Intracellular lncRNAs play a multidimensional role in tumorigenesis and cancer progression by serving as regulators of transcription and translation [23]. Their physiological activity is associated with the regulation of stress response to DNA damage, immune escape, and cell metabolism [24,25,26]. Therefore, the dysregulation of lncRNA expression may arise during tumor formation and aid the development of cancer hallmarks. For instance, during the epithelial–mesenchymal transition (EMT), cancer-associated fibroblasts secrete transforming growth factor β (TGF-β), which in turn increases lncRNA levels. Of them, lncRNA-ATB upregulated the zing finger E box-binding homeobox (ZEB1 and ZEB2), facilitating the EMT and driving distant metastasis in hepatocellular carcinoma [27]. LncRNA MRPL23 antisense RNA1 (MRPL23-AS1) stimulated EMT by enhancing the activity of the enhancer of the zeste homolog 2 (EZH2) protein and its binding on the E-cadherin promoter region in adenoid cystic carcinoma (ACC). Interestingly, MRPL23-AS1-transfected exosomes promoted the expression of VEGF by down regulating E-cadherin in human pulmonary microvascular endothelial cells. Furthermore, the injection of exosomes from MRPL23-AS1-overexpressing cells into immunodeficient mice facilitated ACC lung metastases [3]. The underlying phenomenon may be caused by increased microvascular permeability since the MRPL23-AS1-dependent changes in VEGF-A and E-cadherin expressions were associated with the increased leakage of Evans blue in mouse lungs [3]. A high expression of MRPL23-AS1 in osteosarcoma was also associated with larger, more advanced, and metastatic tumors. Mechanistically, MRPL23-AS1 competed with miR-30b and increased the expression of myosin heavy chain 9 (MYH9), thus activating the Wnt/β-catenin signaling pathway and promoting osteosarcoma progression [4]. LncRNA is also a key driver of cancer cell survival. After extravasation on the metastatic site, cancer cells with a high stemness and self-renewal ability form micrometastases, which constitutes a crucial step in cancer progression [28]. However, the role of lncRNA in the regulation of cancer stemness seems two-sided, as in hepatocellular carcinoma, lncBRM increased the self-renewal of cancer stem cells. At the same time, lncRNA TSLNC8 inhibited the IL-6/STAT3 signaling pathway, inducing tumor suppression [29,30]. Interestingly, genetic and epigenetic alteration of the 11p15.5 region have been associated with Beckwith–Wiedemann and Silver–Russell syndromes, but a direct link to MRPL23 mutation has yet to be found. Further investigation into the precise roles of MRPL23 and MRPL23-AS1, especially their potential interaction in the tumor microenvironment, could provide valuable insights into the mechanisms underlying tumor progression and reveal novel therapeutic targets [3,4].

In summary, although MRPL23 has shown prognostic significance in ccRCC, there is still a substantial gap in understanding its functions and underlying mechanisms. Further research is crucial to determine whether MRPL23 operates through common molecular pathways across different cancer types or if these mechanisms vary depending on the specific type of cancer. Such knowledge could shed light on its role in tumor biology and potentially lead to the identification of novel therapeutic targets. Additionally, exploring the interaction between MRPL23 and MRPL23-AS1 may provide deeper insights into their combined impact on cancer progression and support the development of targeted treatment strategies.

However, certain limitations of the study should be considered. First, the results of our institutional cohort may not be directly comparable with the TCGA data due to differences in methodology and the characteristics of the studied populations. An important limitation is that our cohort data included cN status, while TCGA data referred to pN status, which may affect the interpretation of the results. Nevertheless, our study was conducted with due diligence and precision, providing a solid foundation for future analyses and indicating a direction for further research on the role of MRPL23 in cancer.

## 5. Conclusions

Advancements in biological research highlight MRPL23 as a potential prognostic marker in cancer. This study shows that MRPL23 expression is elevated in ccRCC tissues compared to non-cancerous tissue, with higher levels correlating with poorer patient outcomes. Despite its significance in predicting prognosis, the mechanisms of MRPL23 in cancer progression remain unclear. Further research is essential to understand whether MRPL23 functions through common pathways across cancers or varies by cancer type, which could guide targeted therapies.

## Figures and Tables

**Figure 1 cancers-16-03909-f001:**
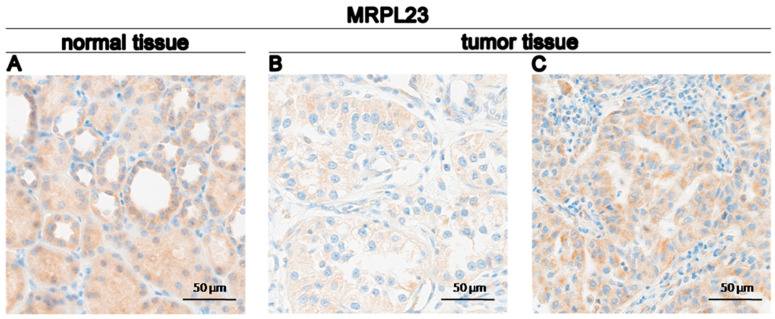
Representative immunohistochemical staining for MRPL23 in ccRCC tissues. (**A**) Normal tissue showing MRPL23 staining. (**B**) Tumor tissue with weak MRPL23 staining. (**C**) Tumor tissue with relatively stronger MRPL23 staining.

**Figure 2 cancers-16-03909-f002:**
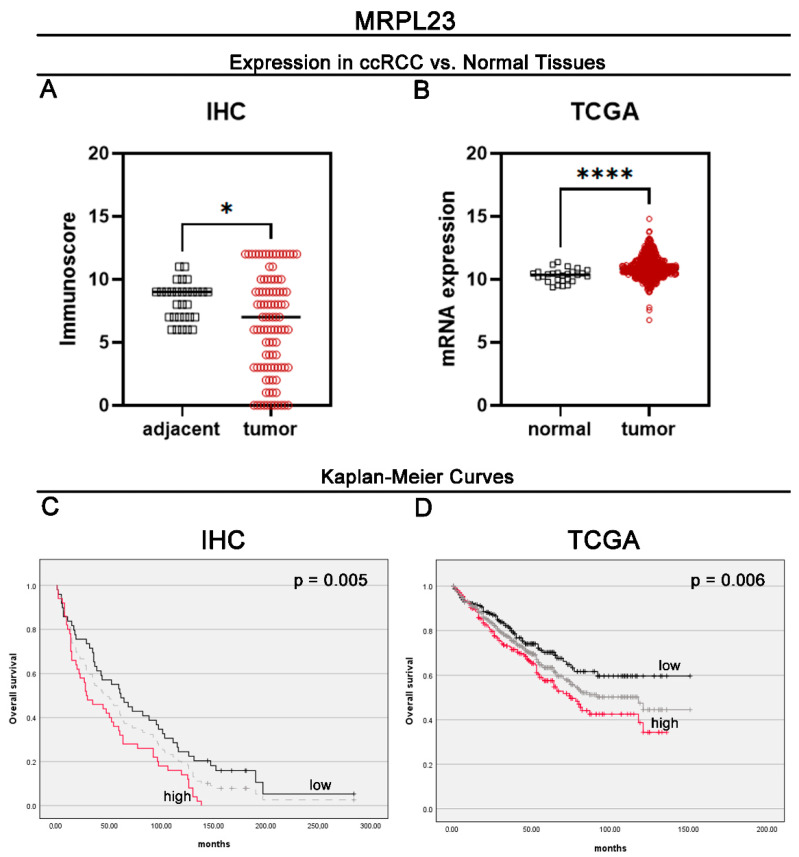
Protein and mRNA expression of MRPL23 in ccRCC. Protein (**A**) and mRNA expression (**B**) of MRPL23 in tumor and adjacent tissues in ccRCC. Kaplan-Meier survival curves and log-rank test for overall survival of ccRCC patients based on MRPL23 protein expression (**C**) and MRPL23 mRNA expression (**D**). * indicates *p* < 0.05, representing statistical significance. **** indicates *p* < 0.0001, representing very high statistical significance.

**Table 1 cancers-16-03909-t001:** MRPL23 protein expression and its relationship with clinicopathological characteristics of clear-cell renal cell carcinoma (ccRCC) patients in our cohort.

Variables	Number (%)	MRPL23		
		High	Low	*p*-Value
		*n* = 50	*n* = 49	
Gender				
Females	31 (31.31%)	17 (54.84%)	14 (45.16%)	0.6658
Males	68 (68.69%)	33 (48.53%)	35 (51.47%)	
Age				
≤65	58 (58.59%)	26 (44.83%)	32 (55.17%)	0.2223
>65	41 (41.41%)	24 (58.54%)	17 (41.46%)	
Grade				
G1	25 (25.25%)	11 (44.00%)	14 (56.00%)	0.1345
G2	64 (64.65%)	31 (48.44%)	33 (51.56%)	
G3	10 (10.10%)	8 (80.00%)	2 (20.00%)	
pT status				
Tx	1 (1.01%)			
T1	28 (28.28%)	16 (57.14%)	12 (42.86%)	0.5286
T2	28 (28.28%)	11 (39.29%)	17 (60.71%)	
T3	40 (40.40%)	22 (55.00%)	18 (45.00%)	
T4	2 (2.02%)	1 (50.00%)	1 (50.00%)	
cN status				
N0	92 (92.93%)	45 (48.91%)	47 (51.09%)	0.436
N1	7 (7.07%)	5 (71.43%)	2 (28.57%)	

**Table 2 cancers-16-03909-t002:** MRPL23 protein expression and its relationship with clinicopathological characteristics of clear-cell renal cell carcinoma (ccRCC) patients in TCGA cohort.

		MRPL23		
Variables		+	−	*p*-Value
		*n* = 240	*n* = 235	
Gender				
Females	163 (34.32%)	70 (42.94%)	93 (57.06%)	0.0203
Males	312 (65.68%)	170 (54.49%)	142 (45.51%)	
Age				
≤60	239 (50.32%)	119 (49.79%)	120 (50.21%)	0.7833
>60	236 (49.68%)	121 (51.27%)	115 (48.74%)	
Grade				
G1	11 (2.32%)	4 (36.36%)	7 (63.64%)	0.0079
G2	203 (42.74%)	86 (42.36%)	117 (57.64%)	
G3	189 (39.79%)	106 (56.08%)	83 (43.92%)	
G4	72 (15.16%)	44 (61.11%)	28 (38.89%)	
pT status				
T1	237 (49.89%)	94 (39.66%)	143 (60.34%)	<0.0001
T2	61 (12.84%)	37 (60.66%)	24 (39.34%)	
T3 and T4	177 (37.26%)	109 (63.74%)	62 (36.26%)	
pN status				
Nx	235 (49.47%)			
N0	225 (47.37%)	109 (48.44%)	116 (51.56%)	0.4334
N1	15 (3.16%)	9 (60.00%)	6 (40.00%)	
Stage				
I	234 (49.26%)	93 (39.74%)	141 (60.26%)	<0.0001
II	50 (10.53%)	26 (52.00%)	24 (48.00%)	
III	119 (25.05%)	75 (63.03%)	44 (36.97%)	
IV	72 (15.16%)	46 (63.89%)	26 (36.11%)	

**Table 3 cancers-16-03909-t003:** Univariate and multivariate Cox proportional hazards models for OS in clear-cell renal cell carcinoma (ccRCC) patients in our cohort.

	Univariate Analysis				Multivariate Analysis			
Variable	HR	95% CI		*p*-Value	HR	95% CI		*p*-Value
MRPL23	1.81	1.18	2.77	0.01	1.66	1.07	2.56	0.02
gender	0.56	0.36	0.87	0.01	0.58	0.37	0.91	0.02
age	1.62	1.07	2.47	0.02	1.27	0.82	1.98	0.28
Grade	2.90	1.48	5.68	0.002	2.53	1.26	5.07	0.01
pT	1.14	0.75	1.73	0.54	-	-	-	-
cN	3.77	1.68	8.47	0.001	3.69	1.62	8.41	0.002

**Table 4 cancers-16-03909-t004:** Univariate and multivariate Cox proportional hazards models for OS in clear-cell renal cell carcinoma (ccRCC) patients in TCGA cohort.

	Univariate Analysis				Multivariate Analysis			
Variable	HR	95% CI		*p*-Value	HR	95% CI		*p*-Value
MRPL23	1.56	1.13	2.15	0.01	1.18	0.85	1.64	0.32
gender	0.95	0.68	1.31	0.73	-	-	-	-
age	1.01	1.00	1.02	0.08	-	-	-	-
Grade	1.36	0.98	1.87	0.06	1.18	0.85	1.63	0.33
pT	3.18	2.31	4.38	<0.0001	-	-	-	-
pN	3.61	1.91	6.82	0.0001	-	-	-	-
TNM stage	3.60	2.59	5.02	<0.0001	3.43	2.44	4.81	<0.0001

## Data Availability

The datasets generated and analyzed during the current study can be obtained from the corresponding author upon reasonable request.

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
