# Peer review of "High Expression of MRPL23 Is Associated with Poor Survival in Clear-Cell Renal Cell Carcinoma"

_cancers, 2024, doi:10.3390/cancers16233909_

Round 1
Reviewer 1 Report
Comments and Suggestions for Authors
The study is interesting and concise, addressing the potential use of MRPL23 as a prognostic marker in clear cell renal carcinoma. The methods applied are appropriate to the research hypothesis, although they primarily consist of immunohistochemical assays and supported by basic bioinformatical-biostatistical analysis.
A fundamental issue in this study is the accurate identification of the biomolecules subjected to the analysis. After multiple readings of the paper and verification of the references, significant uncertainty and confusion remain. This is because the authors refer to long non-coding RNA (lncRNA) in both the introduction and discussion sections, while the symbols used in the study suggest that the research focused on coding mRNA and its associated protein product. The authors must specify clearly whether they are investigating the prognostic role of MRPL23 mRNA and its protein, or MRPL23-AS1 lncRNA, which originates from an approximate region on chromosome 11 but is distinct from MRPL23. It appears that, despite including paragraphs in the introduction about the role of lncRNA in oncogenesis, the RNA expression data presented pertains to coding sequences rather than non-coding sequences. This fundamental issue must be clearly and unequivocally addressed in the paper. In addition to providing the gene symbol, the authors should supply precise accession numbers for genes/RNAS used in the study.
Did the authors consider the option to obtain RNA from the same tissue fragments used to prepare the tissue microarray? This approach would have allowed for better correlation of protein and RNA levels within the same cases. If the analyzed tissue microarrays were obtained from archival material, this should be stated explicitly in the materials and methods section. In its current form, the paper suggests that the specimens were collected specifically for this study, and that RNA samples could have been easily obtained and included in the analysis.
Why were distant metastases and/or staging (e.g., I+II+III vs IV) not included in the model used for Cox regression? Omitting one of the most significant parameters in survival analysis is a mistake and can influence the results of the multivariate analysis.
Minor issues:
Xena database – specify the date of data accession; when using data provided by Xena, the authors should cite the source following the instructions (https://xena.ucsc.edu/cite-us).
Specify the units used by Xena database? RSEM, TPM, or FPKM? It is necessary when cutoff value is defined.
Which cells were evaluated and quantified after IHC? “Reduced in normal tissues compared to ccRCC tissues?” The authors must specify which cells were quantified for their MRPL23 immunoexpression.
Add a scale bar to Fig 1.
Author Response
The responses to the reviews are included in the attachment.

Reviewer 2 Report
Comments and Suggestions for Authors
Conclusion:
In this study, Edyta et al. demonstrated that MRPL23 protein expression decreases in clear cell renal cell carcinoma (ccRCC), while its mRNA levels increase. This increase was associated with clinicopathologic characteristics and disease progression, suggesting that higher MRPL23 protein and mRNA levels might predict poorer survival outcomes. The findings indicated that MRPL23 protein may serve as a potential biomarker for diagnosing ccRCC. However, some issues still require further explanation.
Major comments
1. Table 1 presents various clinicopathological characteristics, including pT and cN status, used to characterize different stages of disease progression. However, some stages differ from those in Table 2, which may lead to misunderstandings. Please apply the TNM system to standardize the clinical characteristics across all three tables for greater clarity and ease of comparison.
2. In the analysis of the TCGA cohort, MRPL23 mRNA levels were significantly elevated in 162 tumor tissues compared to normal non-tumorous tissues. To assess the degree of MRPL23 gene expression, the results may derive from the quantification of mRNA through real-time PCR using patient samples.
3. How should we interpret the differential expression of protein and mRNA levels in clear cell renal cell carcinoma? Has transcriptional regulation been reported in this context?
4. The author mentioned that MRPL23 is predominantly localized in the nucleus, where it enhances the binding of EZH2 and H3K27me3 to the E-cadherin promoter region, leading to its silencing (lines 62-64). In fact, MRPL23 is also located in the mitochondria and cytoplasm (Fig. 1). What are the functions of these different cellular localizations?
5. MRPL23, one of the long non-coding RNAs (lncRNAs), plays a crucial role in gene regulation. The author performed immunohistochemical staining to detect MATR3 using a rabbit polyclonal anti-MRPL23 antibody, which is derived from a coding gene. What is the functional connection between MATR3 protein and the long non-coding RNAs, MRPL23?
6. The text contains several typos and grammatical errors. Please check it more carefully.
Author Response

(The authors gave the same response as above.)

Round 2
Reviewer 2 Report
Comments and Suggestions for Authors
Thank you for all your explanations and answers.